# Measurement of Intermediate Frequency Magnetic Fields Generated by Household Induction Cookers for Epidemiological Studies and Development of an Exposure Estimation Model

**DOI:** 10.3390/ijerph191911912

**Published:** 2022-09-21

**Authors:** Takumi Kitajima, Joachim Schüz, Akemi Morita, Wakaha Ikeda, Hirokazu Tanaka, Kayo Togawa, Esteban C. Gabazza, Masao Taki, Kuniaki Toriyabe, Tomoaki Ikeda, Shigeru Sokejima

**Affiliations:** 1Department of Public Health and Occupational Medicine, Mie University Graduate School of Medicine, Tsu 514-8507, Japan; 2International Agency for Research on Cancer (IARC), WHO, 69372 Lyon, France; 3Epidemiology Centre for Disease Control and Prevention, Mie University Hospital, Tsu 514-8507, Japan; 4Division of Surveillance and Policy Evaluation, National Cancer Center Japan Institute for Cancer Control, Chuo-ku, Tokyo 104-0045, Japan; 5Department of Immunology, Division of Molecular and Experimental Medicine, Mie University Graduate School of Medicine, Tsu 514-8507, Japan; 6Department of Systems Design, Tokyo Metropolitan University, Hachioji 192-0397, Japan; 7Electromagnetic Compatibility Laboratory, National Institute of Information and Communications Technology, Koganei 184-0015, Japan; 8Department of Obstetrics and Gynecology, Mie University Graduate School of Medicine, Tsu 514-8507, Japan

**Keywords:** intermediate frequency, magnetic field, induction cooker, estimation model, questionnaire, measurements

## Abstract

Introduction: Exposure assessment of intermediate frequency (IF) electromagnetic fields (EMFs) is difficult and epidemiological studies are limited. In the present study, we aimed to estimate the exposure of pregnant women to IF-EMFs generated by induction cookers in the household using a questionnaire and discussed its applicability to epidemiological studies. Method: Two main home-visit surveys were conducted: a Phase 1 survey to develop an estimation model and a Phase 2 survey to validate the model. The estimation model included the following variables: wattage, cookware diameter, and distance from the hob center (center of the stove). Four models were constructed to determine the importance of each variable and the general applicability for epidemiological studies. In addition, estimated exposure values were calculated based on the Phase 2 survey questionnaire responses and compared with the actual measured values using the Spearman rank correlation coefficient. Result: The average value of the magnetic field measured in the Phase 1 survey was 0.23 μT (variance: 0.13) at a horizontal distance of 30 cm at the height of the cooking table. The highest validity model was inputted distance from the hob center to the body surface that is variable (correlation coefficient = 0.54, 95% confidence interval: 0.22–0.75). No clear differences were identified in the correlation coefficients for each model (z-value: 0.09–0.18, *p*-value: 0.86–0.93). Discussion and Conclusions: No differences were found in the validity of the four models. This could be due to the biased wattage of the validation population, and for versatility it would be preferable to use three variables (distance, wattage, and estimation using the diameter of the cookware) whenever possible. To our knowledge, this is the first systematic measurement of magnetic fields generated by more than 70 induction cookers in a real household environment. This study will contribute to finding dose–response relationships in epidemiological studies of intermediate-frequency exposure without the use of instrumentation. One of the limitations of this study is it estimates instantaneous exposure in place during cooking only.

## 1. Introduction

Today, electric home appliances benefit us at home. However, the use of household appliances is associated with increased exposure to electric and magnetic fields in our bodies. Guidelines have been established by the International Commission on Non-Ionizing Radiation Protection (ICNIRP) to prevent adverse effects on health caused by electrical and nerve stimulation after exposure to electromagnetic fields. Manufacturers evaluate the safety of their products based on these guidelines.

Electromagnetic fields of various frequencies exist in our surroundings, and it is believed that different frequencies have different effects. Currently, epidemiological studies of electromagnetic field exposure in any frequency band are often of unclear relevance due to evaluation difficulties and biases. The frequency generally used to supply electricity to homes (power frequency (PF)) is extremely low frequency (ELF), 50 Hz or 60 Hz. Many studies have pointed out the health effects of electromagnetic fields (ELF) in this frequency range on humans [1,2,3,4,5,6,7]. The International Agency for Research on Cancer (IARC) evaluated the carcinogenic effects of extremely low frequency electromagnetic fields (ELF-EMF). It categorized them in Group 2B on the list of carcinogenic hazard identification [8]. Studies have also been conducted on radio frequency (RF; 10 MHz to 300 GHz), a frequency band used in wireless communications such as cellular phones [9,10,11,12,13]. This RF belongs to Group 2B on the carcinogenic hazard identification of IARC [14]. Many in vivo and in vitro studies have also been conducted on exposure to magnetic fields in the intermediate frequency (IF; 300 Hz to 10 MHz) range [15,16,17,18,19,20,21,22,23]. Most reports showed no effect after exposure to IF magnetic fields, although some studies have shown an increase in oxidative stress markers [20]. There are many unknowns regarding the mechanisms of biological effects of IF-EMF exposure, and results have been inconsistent. Therefore, there is an urgent need to accumulate further research in this area, including epidemiological studies. Exploratory studies revealed that induction heating equipment is the most common source of exposure [24,25]. However, there is a lack of epidemiological research on the human health effects of the IF band (e.g., wireless power transmission). The application of the IF band will significantly expand in the coming future.

The main focus of the present study is the induction cooker, the most popular induction heating device in the household [24]. We are currently conducting a study, “Perinatal Cohort Study,” evaluating the health effects on the children of pregnant women who have been exposed to IF electromagnetic fields while using household devices. Since it is not feasible to evaluate exposure to a magnetic field in the more than 1000 women enrolled in the cohort, in this study we aim to estimate the exposure of pregnant women to the magnetic field generated by induction cookers using a questionnaire and discuss the method’s feasibility. Our purpose is to develop and validate a model that describes the magnitude of the magnetic field generated by induction cookers in the home and to estimate the IF exposure of pregnant women while cooking with induction cookers. This study will contribute to finding dose–response relationships in epidemiological studies of intermediate-frequency exposure without the use of instrumentation.

## 2. Materials and Methods

### 2.1. Study Design and Participants

These surveys were conducted in two phases. The Phase 1 survey was conducted as a pilot study in 2018 to collect data for model building. An internet approach was used to survey 1014 women in Mie prefecture Japan about using electronic devices at home. In addition, home visits were made to 50 households that had agreed to have their household IF magnetic field environment measured. Forty-five households and participants were included in the model building, excluding five participants that had gas stoves in the Phase 1 survey. For the Phase 2 survey, intended to provide data for model validation, 30 of the 206 Perinatal Cohort members who agreed to participate in the household environment survey were randomly selected. The Perinatal Cohort Study used a self-administered questionnaire and a questionnaire completed by the attending researchers. The Appendix A show a partially excerpted questionnaire.

The use of the induction cooker, the distance from the hob center (center of the stove) to the body surface in the usual cooking posture, and the diameter of the cookware were self-reported by the participant. Moreover, stature (from the physician’s questionnaire) was used. The participants were instructed to specify the measurement position using a diagram or a measuring tape enclosed with the questionnaire to obtain accurate values of the distance and diameter.

This study was conducted in accordance with the Ethical Guidelines for Medical and Biological Research Involving Human Subjects, and the project protocol was approved by the Clinical Research Ethics Review The project protocol was approved by the Clinical Research Ethics Review Committee of Mie University Hospital (approval numbers: U2018-009, U2019-027).

### 2.2. Measurement Tools and Setting

A Narda EHP-50F was used for the measurements (Narda Safety Test Solutions, Pfullingen, Germany). This measurement device can measure magnetic and electric field strength in three-dimensional directions and has an internal 1024-point fast Fourier transform (FFT) function. The measurable frequency range is 1 Hz to 400 kHz. The values used in this study were in the range of 1 kHz to 400 kHz. The magnetic field root mean square values (wide band values) in the 4.883 k–400 kHz range were used to calculate the values. The devices were calibrated at the time of shipment by the manufacturer, and the maximum uncertainty in the 400 kHz magnetic field measurement range was 3%.

### 2.3. Measurement Procedures

#### 2.3.1. Measuring the Positional Relationship to the Exposure Source

In estimating the positional relationship between the human body and the source of exposure, it is necessary to determine a measurement index. Since the location of the uterus is difficult to identify due to individual differences and shifts in body position, the height of the superior iliac crest identified by palpation was used as an alternative indicator. The iliac crest height was measured with a measuring tape under the guidance of the physician. The participants self-reported assuming their usual cooking position in front of the induction cooker. Moreover, an anthropometer (a large caliper-like instrument) was used to measure the horizontal distance from the edge of the induction cooker to the body surface and from the hob center to the body surface.

#### 2.3.2. Measuring the Magnetic Field of Induction Cookers in Households

For the most frequently used hob, a grid of 10 cm increments was established along with the extension of the center of the hob. The height of the cooktop was taken as a reference, *h* = 0, and measurements were done up and down to 20 cm. In the horizontal direction, the sides of the cooktop were set to a distance of *d* = 0, and measurements were done up to *d* = 30 cm. A schematic diagram of the measurement points is shown in Figure 1. At the time of measurement, the water was set to maximum output and heated with each household’s most frequently used cookware. The cookware was placed in the center of the stove. The heating time was set to 1 min.

### 2.4. Estimation Model Construction and Validation

The mechanism by which an induction cooker heats the bottom of the cookware uses the intermediate frequency magnetic field generated by the heating coil to generate an induction current in the metal cookware. Joule heat generated by the electric resistance of the cookware itself is used. Theoretically, the magnetic field at an arbitrary point can be determined by the magnitude of the current, the distance from the current flow path, and the magnetic permeability. For example, a law of physics is Biot–Savart’s law. However, to calculate the magnetic field generated by an induction cooker as precisely as in this case, it is necessary to know all the paths of the current flowing in the device. This means disassembling the equipment under investigation and observing a complex system, which is not realistic in an on-site survey. It is also difficult to measure the induced currents generated in the bottom of the cookware. A simple method to determine the magnetic field generated by induction cookers was based on the literature by Yamasaki et al. [26]. The purpose of this paper is to estimate the magnetic field generated by an appliance by considering the appliance as a loop coil, which is the source of exposure to the magnetic field. However, not all users cook under the same conditions in real life. For example, the diameter of the cooking utensils usually used and the heating capacity of the equipment are expected to vary from household to household. These are variables associated with the current. The model in this paper is linearly populated with these variables in the denominator for adjustment. Concerning the variables included in the model, wattage is expressed as the product of the resistance of the load and the current squared, thus taking the square root from Ohm’s law. Moreover, when feeding the cookware diameter into the model, the square of the radius was used as a variable, as leakage magnetic fields and loading were considered to be related to the area of the bottom of the cookware.

In epidemiological studies, these variables are not always available from questionnaires. Therefore, it is necessary to compare the model fit for each combination of variables used and to compare the contribution of each variable to the IF-EMF estimate and its applicability to epidemiological studies. In this study, four models were constructed. Models 1 to 4 correspond to Equations (1)–(4). Model 1 includes wattage W, cookware diameter $\phi_{c}$, and distance; Model 2 includes wattage W and distance variables only; Model 3 includes cookware diameter $\phi_{c}$ and distance variables only; and Model 4 includes only distance variables. The current term coefficient is $\beta_{1}$, the cookware diameter adjustment term factor is $\beta_{2}$, and the intercept adjustment term is ε. The height *h* was calculated by using the height of the superior border of the iliac crest estimated from height, assuming the height of the cooking table to be 85.0 cm, and finding the difference between the two. The origin of the cooker height is discussed in the Results section. The difference between the height of the superior margin of the iliac crest and the typical height of the cooking table, 85.0 cm, is defined as *h*. The height of the iliac crest is determined from the stature. Equation (5) shows how the height *h* was obtained. Magnetic field measurement data from 45 induction cookers (sample size *n* = 45 target households × 20 measurement points = 900) were obtained in the Phase 1 survey. Moreover, $\beta_{1}$, $\beta_{2}$, and ε were determined. Non-linear regression was used to determine the coefficients, and the least-squares method fit the coefficients to the measured data.

The validity of each estimation model was assessed using the data from Phase 2 surveys. We first estimated exposure values using the estimation models constructed in Phase 1 and the questionnaire responses obtained in the Phase 2 survey. We then calculated the rank correlation coefficient between the estimated and measured values in 30 participants from Phase 2 using the Spearman rank correlation coefficient. R version 3.6.0 (R Core Team (26 April 2019). R: A language and environment for statistical computing. R Foundation for Statistical Computing, Vienna, Austria, http://www.R-project.org/) was used for model construction and all statistical analyses.
(1)$$\mathrm{Model}\;1\;B_{1}=\frac{\beta_{1}\mu \sqrt{W}+\beta_{2}{\left(\frac{\phi_{c}}{2}\right)}^{2}+\varepsilon }{4\pi {\left(\sqrt{d_{h}^{2}+h^{2}}\right)}^{3}}$$
(2)$$\mathrm{Model}\;2\;B_{2}=\frac{\beta_{1}\mu \sqrt{W}+\varepsilon }{4\pi {\left(\sqrt{d_{h}^{2}+h^{2}}\right)}^{3}}$$
(3)$$\mathrm{Model}\;3\;B_{3}=\frac{\beta_{2}{\left(\frac{\phi_{c}}{2}\right)}^{2}+\varepsilon }{4\pi {\left(\sqrt{d_{h}^{2}+h^{2}}\right)}^{3}}$$
(4)$$\mathrm{Model}\;4\;B_{4}=\frac{\varepsilon }{4\pi {\left(\sqrt{d_{h}^{2}+h^{2}}\right)}^{3}}$$
$B_{n}:\;Predicted\;Magnetic\;field\left(\mu T\right),$$\mu :\;Magnetic\;Transmittance\left(=1.257\times 10^{-6}\right),\;W:Wattage\left(W\right),$$\phi_{c}:\;Diameter\;of\;cook\;tools\left(m\right),\;d_{h}:\;Horizontal\;distance\;from\;Hob\;center\left(m\right),$$h:\;Height\;from\;cooking\;table\;to\;superior\;margin\;of\;iliac\;crest\left(m\right)$
(5)$$height\;of\;\mathrm{iliac}\;\mathrm{crest}\;\left(m\right)=\frac{Stature\;\left(\mathrm{cm}\right)\;\times \;0.6935\;-\;16.997}{100}\;h=height\;of\;\mathrm{iliac}\;\mathrm{crest}\;\left(m\right)-0.85$$

## 3. Results

### 3.1. Distribution of Positional Relationship between the Participants and Induction Cooker

Figure 2 shows the relationship between the iliac crest’s height and the female participants’ stature. The average height was 158.3 cm, and the average height of the iliac crest was 92.8 cm. There was a proportional relationship between the height of the iliac crest and stature.

Figure 3 shows the distance relationship between the induction cooker and the female participant: (a) shows the distance from the edge of the cooking table to the body’s surface in the usual cooking position, and (b) shows the distance from the center of the hob of the induction cooker. The average distance in (a) was 15.00 cm (standard deviation (SD): 11.19) and in (b) was 38.98 cm (SD: 11.68). A total of 98% of the 45 participants were cooking standing within 30 cm of the edge. This means that the body is located closer than the measurement point defined in IEC 62233 [27]. The average cooking table height in the target households was 85.04 cm (SD: 0.89). This is the height specified by the Japanese Industrial Standards (JIS) and is the standard for most households in Japan.

### 3.2. Characteristics of Induction Cookers and Distribution of Measured Magnetic Fields

The distribution of the fundamental frequency is shown in Figure 4. The fundamental frequency ranged from 20 k to 30 kHz, the optimum frequency for heating iron and stainless steel. However, heating by equipment compatible with aluminum pots used a frequency of 75 kHz. The highest value for the measured magnetic flux density was obtained at a distance closest to the enclosure in the horizontal plane of the cooking table (measurement grid: *d* = 0, *h* = 0), with an average value of 3.864 μT (SD: 3.311). Table 1 shows the measured magnetic flux densities distribution for the measurement grid. Magnetic fields were measured at the actual cooking position, at the height of the superior iliac crest, during the Phase 2 survey and averaged 1.186 μT (SD: 0.908).

### 3.3. Estimated Model Coefficients and Validation

Table 2 shows the coefficients of the estimated equations. For Model 1, there were significant contributions for the current term involving wattage and the cookware diameter adjustment term. For Model 2, there were significant contributions for the current term involving wattage and the cookware diameter adjustment term. For Model 2, wattage was a significant contributor; for Model 3, both cookware diameter and intercept adjustment terms were significant contributors; and for Model 4, only one variable, the intercept adjustment term, was significant. The model with the lowest AIC was Model 1.

As a validation, the correlations between the measured magnetic fields generated by the induction cooker and the magnetic fields estimated from each model were calculated. Table 3 shows the correlation between measurement value and estimated value. All correlation coefficients from Mode 1 to Model 4 were statistically significant and ranged between 0.47 and 0.54. No clear differences were identified in the correlation coefficients for each model (z-value: 0.09–0.18, *p*-value: 0.86–0.93). Model 4 showed a slightly better performance than the other three, as it resulted in the highest validity (r = 0.54, 95% CI: 0.22–0.75). Figure 5 shows a scatter plot of the estimated and measured values calculated from the model.

## 4. Discussion

### 4.1. Position of Participants in Relation to the Induction Cooker

The distance of the measuring device from the edge of the induction cooker was 30 cm, as specified by IEC 62233 [28]. The appropriateness of this distance has been pointed out in the past [29]. The results of the present study suggest that many women in Mie Prefecture, Japan, stand within at least 30 cm of the edge of the cooktop and cook at an average distance of 15 cm; this totals a distance of about 35 cm from the center of the hob, with a flux density distribution in the range of 0.2 to 2.0 μT. It is unrealistic to always stand still before an induction cooker. Investigation of the precise actual time spent before the cooker is very important. Furthermore, spontaneous exposure during the cooking time should be used as an indicator of exposure in epidemiological studies.

There were no cases where the exposure values exceeded the ICNIRP public exposure guideline (27.0 μT in all frequency bands from 3 kHz to 10 MHz). The IEC 62233 was met. However, the possibility that the magnetic fields were diffused, exceeding the guidelines, cannot be ruled out.

### 4.2. About Measurement of Magnetic Fields

We compared the values obtained in our present investigation with values reported in a previous study that measured 16 induction cookers in Switzerland. At similar points, our measurements averaged 3.864 μT (SD: 3.311), whereas those of Christ et al. [29] averaged 21.44 μT (SD: 14.99), a value six times higher. The possible reason for this is that the distance from the center of the hob to the edge of the cooktop was shorter than the distance from the center of the hob to the edge of the cooktop. It averaged 17.9 cm (SD: 5.97) in the product measured by Christ et al., whereas it averaged 23.9 cm (SD: 3.38) in our study. This is thought to be due to the shorter distance from the center of the hob to the edge than the Japanese products. The effect of wattage was not described in the study by Christ et al.

The trend of the measurement results shows that as the horizontal distance d increases, the measured values’ dispersion tends to be smaller than in the near vicinity of the equipment. At any height, the upper quartile was about 2 to 3 times larger than the lower quartile at a horizontal distance of *d* = 30 cm, which is generally measured. In other words, it is desirable to use cutoff values with high specificity when categorizing by exposure in epidemiological studies.

### 4.3. About Estimation Models

Validation results showed that the correlation coefficients for each model ranged from 0.47 to 0.54; hence, the models performed with a relative similarity. Moreover, the selection of variables was considered appropriate since no model significantly lacked in validity and the parameters for each model also contributed significantly. Observing the plot characteristics in Figure 5, larger exposures may result in underestimation. In addition, estimates below 1 μT are overestimates. This may be attributed to the large variety of magnetic fields among devices in the vicinity of the cooktop edge. One possible reason for this is that there is a large variation in magnetic fields due to individual differences in the vicinity of induction cookers (areas where the difference between estimated and measured values is large). Although this estimation model uses distance, wattage, and cookware diameter as variables, which are information that can be obtained from the questionnaire, the possibility that latent factors may be included cannot be denied.

Although the model’s explanatory power was expected to improve with the input of all selected variables in Model 1, the difference was not confirmed. One potential reason for this is that the distribution of wattages in the data used for model building differs from the distribution of wattages in the data for validation; the data for model building obtained from the 2018 survey were sampled to avoid the duplication of models to accommodate magnetic field estimation for a wide range of models, and the wattage values were distributed over a range of 1.5 kW to 3 kW. In contrast, the validation data collected in the homes of the Phase 2 survey in 2021 showed that the wattage distribution was biased toward 3 kW, suggesting that the input of the wattage variable caused the estimation error (F = 2.185, *p*-value = 0.0103). The current flowing through the heating coil and pot also varies with the voltage output from the inverter inside the induction cooker. Therefore, even if the wattage of the induction cooker is the same, the strength of the generated magnetic field may vary depending on the model of the cooker. Furthermore, the effect of the change in the distance on the exposure is significantly larger than any other variable due to the nature of the equation and thus has less impact on the estimated value than the wattage or the diameter of the cooker. Therefore, no clear difference in the estimation validity of the four models emerged.

Assessing exposure in a large-scale epidemiological study requires an easy questionnaire coupled with a valid exposure estimation method. However, especially in this study, these items may be relatively difficult to answer because there were occasions when study participants did not know the model and wattage of their cookers and had to refer to catalogs or instruction manuals for confirmation. In addition, there are more than 1000 induction cooker types distributed in Japan, and there is no database of specifications for each type of cooker. Therefore, identifying the wattage from the appliance model name was also a laborious task. However, from the population of this validation, no change in validity can be confirmed due to differences in wattage. Therefore, the best policy is to adopt Model 1, which has the lowest AIC, as much as possible.

### 4.4. Strengths of This Study

Although there are studies on the IF band focusing on induction cookers and performing actual measurements, the targets for the measurements have been limited to a few models [26]. In addition, previous case-control studies have not assessed quantitative exposure [30,31,32]. This is the first study in which more than 70 actual measurements and usage conditions were investigated in an actual household environment. In many cases, exposure was assessed based on whether or not the equipment was used. The present study may help investigate dose–response relationships.

### 4.5. Limitations

The output values in this model consider only instantaneous exposure and do not consider movement during cooking or adjustments in heat output. However, based on previous studies. it may be possible to assume that the magnetic field varies linearly with the scale of the heat power. In addition, the validity of this model is guaranteed only for Japanese houses. Outside of Japan, it is possible to apply this model by confirming that the values of the magnetic fields generated do not differ depending on the voltage used or the type of cooking device used, and then re-describing the body shape and height of the woman and the dimensions of the cooking table.

In addition, this estimation model is dedicated to estimating exposure when induction cookers are used and does not consider other equipment. Another source of exposure to mid-frequency electromagnetic fields is an anti-theft gate (EAS gate), which differs from an induction cooker in that it is not certain whether the exposed person is aware of passing through the anti-theft gate or not. Fortunately, the electromagnetic field output of induction cookers is controlled by the exposed person, so it is possible to know the cooking time and heat power. In the future, it would be desirable to have the examinees wear a portable exposure meter and keep a daily diary to record their daily activities.

## 5. Conclusions

Our findings suggest that distance from the stove’s center, wattage, and cookware diameter could largely account for exposure from household induction cooktops. Validation results showed that the model that did not include wattage but only the diameter of the cookware and the distance between the body and the enclosure could estimate the exposure with the highest validity. In conclusion, the distance between the body and the source of exposure is the most important information and the minimum requirement for estimating exposure to the iliac crest. However, because of bias in the model characteristics of the population for which validation was conducted, the use of Model 1 is recommended if all the values of the variables presented in this study can be obtained by questionnaire. This work estimates exposure in a fixed position during cooking. Therefore, for the exposure estimate of epidemiological studies, information on the cooking duration has to be included.

## Figures and Tables

**Figure 1 ijerph-19-11912-f001:**
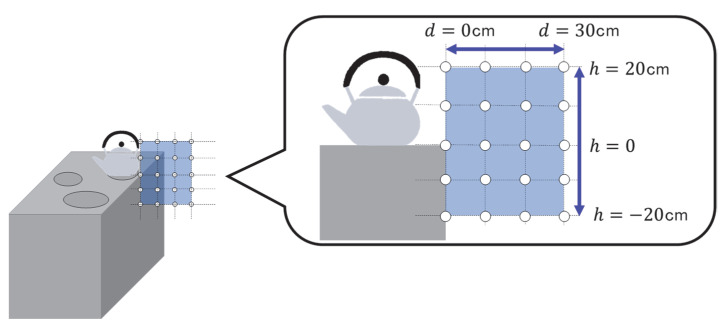
Schematic diagram of magnetic field measurement points in Phase 1 household environment measurement, based on the measurement method of IEC 62233; the number of measurement points was set at 20 to provide a measurement point where a woman’s pelvis fits and to take into account various cooking postures. The enclosure plane is defined as the height reference (*h* = 0), and the edge of the enclosure (Edge) is defined as *d* = 0.

**Figure 2 ijerph-19-11912-f002:**
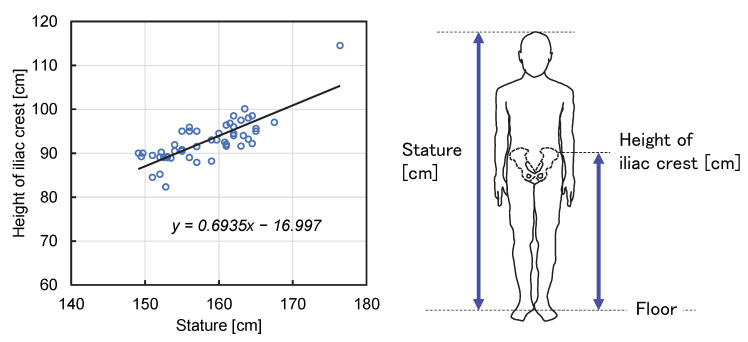
Stature and superior iliac crest height relationship for female participants. The two values are proportional.

**Figure 3 ijerph-19-11912-f003:**
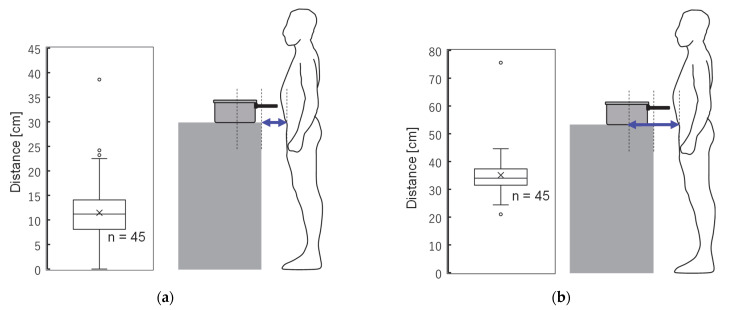
Distance distribution between cooking device and body surface in the participants’ usual posture (cm): (**a**) distance of body surface from cooking table edge and (**b**) distance of body surface from hob. For the center at distance (**a**) the average was 11.2 cm and at distance (**b**) it was 35.0 cm.

**Figure 4 ijerph-19-11912-f004:**
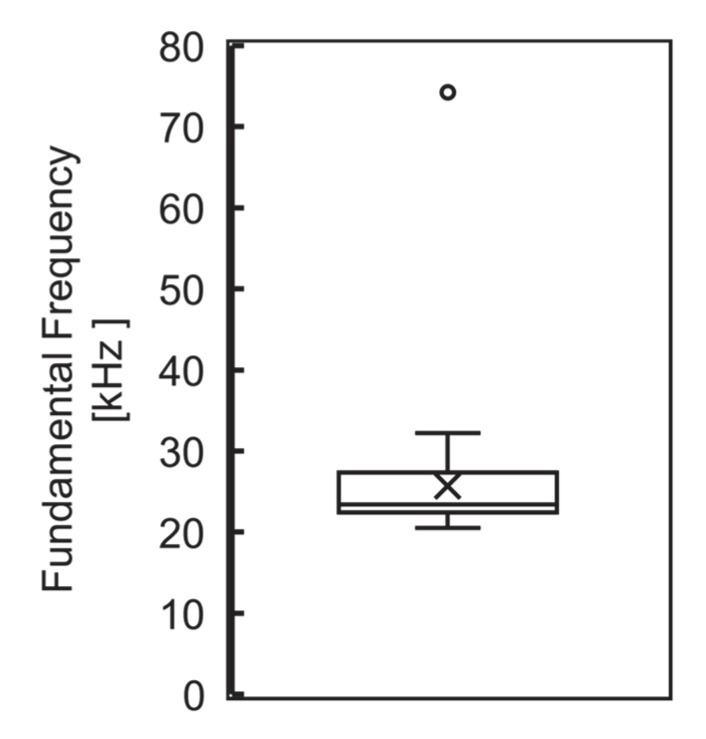
Distribution of fundamental frequencies (*n* = 45): 20 kHz to 30 kHz, which is suitable for heating iron and stainless steel. A high frequency plot was recorded for heating aluminum pots; none of them deviated from the intermediate frequency range.

**Figure 5 ijerph-19-11912-f005:**
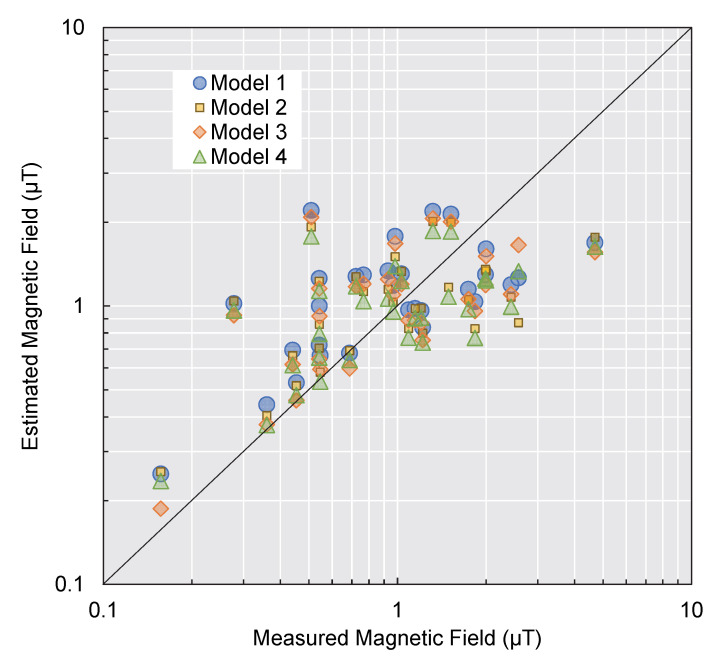
Comparison of magnetic field estimated from questionnaire responses and actual measured magnetic field. The closer the plot is to the black shaded line in this figure, the more accurate the estimation.

**Table 1 ijerph-19-11912-t001:** Measurements of magnetic field around household induction cooker.

Prove Position (cm)			Measured Magnetic Field (μT)						
*h*	*d*	*n*	Mean (SD)		GM (GSD)		1st Qu.	Median	3rd Qu.
20	0	45	1.93	(1.82)	1.36	(2.29)	0.65	1.22	2.33
	10	45	0.81	(0.61)	0.63	(2.00)	0.39	0.63	1.04
	20	45	0.36	(0.23)	0.31	(1.77)	0.21	0.33	0.412
	30	45	0.20	(0.12)	0.17	(1.66)	0.13	0.17	0.24
10	0	45	2.88	(2.18)	2.22	(2.10)	1.41	2.03	3.83
	10	45	0.81	(0.45)	0.71	(1.71)	0.55	0.69	0.99
	20	45	0.35	(0.19)	0.31	(1.70)	0.22	0.31	0.445
	30	45	0.20	(0.10)	0.17	(1.64)	0.12	0.19	0.25
0	0	45	3.86	(3.31)	2.88	(2.24)	1.77	2.96	5.11
	10	45	1.06	(0.70)	0.84	(2.07)	0.59	0.94	1.39
	20	45	0.43	(0.25)	0.36	(1.96)	0.25	0.41	0.60
	30	45	0.23	(0.13)	0.19	(1.87)	0.11	0.21	0.31
−10	0	45	2.154	(2.010)	1.39	(2.84)	0.74	1.64	2.55
	10	45	0.917	(0.746)	0.68	(2.25)	0.36	0.80	1.15
	20	45	0.436	(0.275)	0.35	(1.99)	0.20	0.38	0.63
	30	45	0.226	(0.144)	0.19	(1.87)	0.11	0.21	0.32
−20	0	45	2.092	(2.059)	1.20	(3.27)	0.46	1.52	2.93
	10	45	0.732	(0.602)	0.52	(2.41)	0.23	0.59	1.05
	20	45	0.358	(0.270)	0.276	(2.09)	0.15	0.29	0.53
	30	45	0.198	(0.132)	0.162	(1.90)	0.10	0.18	0.26

SD: standard deviation; GM: geometric mean; GSD: geometric standard deviation; 1st Qu.: first quartile; 3rd Qu.: third quartile. We provided magnetic flux density values; measurement range of the prove: 100 μT range at 400 kHz span.

**Table 2 ijerph-19-11912-t002:** Coefficients of the calculated estimation equations.

	Parameter	Value	t-Value	*p*-Value	AIC
Model 1	$\beta_{1}$ (Current term)	10,501.2	3.60	<0.001	2726.3
	$\beta_{2}$ (Cookware diameter adjustment term)	−28.8	−6.88	<0.001	
	ε (Intercept adjustment term)	0.173	0.875	0.382	
Model 2	$\beta_{1}$ (Current term)	12,160.7	4.07	<0.001	2770.5
	ε (Intercept adjustment term)	−0.216	−1.11	0.266	
Model 3	$\beta_{2}$ (Cookware diameter adjustment term)	−30.0	−7.16	<0.001	2737.2
	ε (Intercept adjustment term)	0.878	19.8	<0.001	
Model 4	ε (Intercept adjustment term)	0.573	36.6	<0.001	2785.1

AIC: Akaike’s information criterion.

**Table 3 ijerph-19-11912-t003:** Comparison between prediction value and measurement value (*n* = 30).

	Correlation Coefficient	(95% CI)		*p*-Value
Model 1	0.500	(0.171)	(0.730)	<0.001
Model 2	0.472	(0.134)	(0.711)	<0.001
Model 3	0.521	(0.198)	(0.742)	<0.001
Model 4	0.540	(0.223)	(0.754)	<0.001

## Data Availability

Not applicable.

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
