# Peer review of "Measurement of Intermediate Frequency Magnetic Fields Generated by Household Induction Cookers for Epidemiological Studies and Development of an Exposure Estimation Model"

_ijerph, 2022, doi:10.3390/ijerph191911912_

Round 1
Reviewer 1 Report
This paper develops the models that estimate the magnitude of the magnetic field close to the IH cookers. It is useful for people’s health care.
I have some comments below;
1)
Please emphasize the novelty of this research in the Abstract and Introduction. This paper shows the [Strengths of This Study] in Sec. 4.4. It is better to state the novelty in the abstract too.
And the abstract should not include undefined parameters. This abstract has [r] without definition.
2)
This paper has many parameters and abbreviations. Please make a nomenclature for readers.
3)
This paper suggests four models as Eqs. (1)-(4). Please show the grounds for proposing these equations. These equations are key points of this paper.
In these equations, phi-c is introduced. However, phi-p is explained in L188. Which is correct in this paper?
4)
Where is the source for Eq. (5)?
5)
What is N in Fig. 3. It is a sample size, I think. However, it is shown in n or N in the text.
The vertical axis in the graph should have unit [cm].
6)
Figure 4 has Japanese characters. Please revise the graph.
7)
Some parameters in Tables 1 and 2 have units. Especially, epsilon has a unit [m^2] based on Eq. (1) etc.
8)
Please check the column in Table 3.
What is [r] in the table. Please define the parameter.
9)
In Fig. 5, the magnetic field is estimated lower than the measured values at the measured value is more than 1. And it is estimated higher than the measured values at less than 1. I am interested in the reason.
10)
The references 3 and 23 do not have enough information.
11)
I can not proofread English because I am not a native English speaker. I recommend you to take a proofread by a native English speaker if you have not proofread in English yet.
---
Author Response
Dear Reviewr 1
Thank you for taking the time in your busy schedule to peer review. Your many suggestions have made it possible for us to revise the manuscript in light of the considerations for our readers.
Please see the attachment.

Reviewer 2 Report
Referee report on “Measurement of Intermediate Frequency Magnetic Fields Generated by Household Induction Cookers for Epidemiological Studies and Development of an Exposure Estimation Model” by Takumi Kitajima et al.
The article is devoted to an interesting problem estimation the exposure of pregnant women to IF-EMF generated by induction cookers in the household and should be published after some edits.
1. In the introduction, it is desirable to describe the mechanism of how a magnetic field can interact with living tissues.
2. It is necessary to describe in more detail the physics of the radiation of a magnetic field by an induction cooker. You can make the description quite simple, but understandable to specialists from other related specialties.
3. The relevance of the work and compliance with the current state of research requires more specific confirmation. Among the first 25 references, only one is from 2019, the rest are quite old.
4. Reference [3]. No publication year.
Author Response
Dear Reviewer 2
Thank you for taking the time in your busy schedule to peer review. Your many suggestions have made it possible for us to revise the manuscript in light of the considerations for our readers.
Please see the attachment.

Round 2
Reviewer 1 Report
The paper is revised based on the reviewer comments. The revised paper is sufficient for publication.
---
Reviewer 2 Report
The authors have significantly improved their original manuscript, which now can be recommended for publication.